# OpenReview forum: "ReliabilityRAG: Effective and Provably Robust Defense for RAG-based Web-Search"
_NeurIPS.cc/2025/Conference — NeurIPS 2025 poster_

### Official Review · Reviewer_Xjz6 · 2025-06-30

**Clarity:** 3
**Significance:** 3
**Originality:** 3
**Rating:** 4
**Confidence:** 4

**Summary:**

This paper proposes ReliabilityRAG, a defense framework for Retrieval-Augmented Generation (RAG) systems that leverages document reliability signals to improve robustness against retrieval-time adversarial attacks. The approach combines a graph-theoretic method based on Maximum Independent Set (MIS) with a scalable weighted sampling framework, and provides both theoretical guarantees and empirical evidence of improved robustness and utility.

**Questions:**

1. How robust is the approach to NLI degradation? If the NLI model used for contradiction detection underperforms—e.g., on non-English or technical content—how significantly does this affect the reliability of the MIS defense?

2. Could the sampling parameters be learned or adapted? Is it feasible to tune the sampling distribution or context size in a data-driven manner, rather than fixing γ, T, and m heuristically?

3. Can the MIS selection be integrated with document reranking? For example, would combining MIS with re-ranking via learned relevance scores further enhance robustness or utility?

4. What happens when no consistent majority exists? In cases where most retrieved documents are noisy, off-topic, or benignly contradictory, how does the system avoid excessive pruning or failure?

**Ethical Concerns:**

["NO or VERY MINOR ethics concerns only"]

**Final Justification:**

The authors’ rebuttal and clarifications resolved some of my concerns, while a few issues remain; my final score reflects this balance.

**Limitations:**

The paper discusses limitations in Appendix D.3, but a few areas warrant further elaboration:

1. The approach presumes the existence of a coherent, contradiction-free majority of retrieved documents. This assumption may not hold in highly ambiguous or multi-perspective queries, or when the retriever surfaces highly diverse sources.

2. The scalability of the full MIS procedure is limited to relatively small k (e.g., ≤ 20). While the sampling extension alleviates this, its effectiveness is tied to careful parameter tuning, which may not generalize across domains or tasks.

3. The reliance on search engine ranking as a proxy for reliability may not transfer to settings outside of web search, such as academic corpora, social media, or enterprise knowledge bases.

4. The filtering mechanism (based on “I don’t know” responses) introduces dependence on LLM behavior, which could be brittle or model-specific.

**Paper Formatting Concerns:**

No.

**Quality:**

3

**Strengths And Weaknesses:**

# Strengths:

1) Well-motivated and timely problem: The paper tackles a concrete and increasingly important vulnerability in retrieval-augmented generation (RAG) systems—namely, the corruption of retrieved documents via prompt injection and corpus poisoning. Framing this challenge in the context of search-based AI applications (e.g., Google’s AI Overviews) grounds the work in a highly relevant and practical setting.

2) Principled algorithmic contribution: The authors introduce a graph-theoretic defense mechanism based on finding a maximum independent set (MIS) over contradiction relationships among retrieved documents. This approach is both intuitive and theoretically grounded, with robustness guarantees under natural assumptions.

3) Theoretical rigor: The authors formalize a threat model and provide provable robustness guarantees for their algorithms. These are nontrivial results that bolster the paper’s credibility and potential impact.

# Weaknesses:

1) Reliance on NLI performance: The MIS approach depends critically on a Natural Language Inference (NLI) model to detect contradictions between isolated LLM responses. While the authors use a strong pre-trained model, performance in open-domain or domain-specific contexts may vary considerably, which could undermine the robustness guarantee in practice.

2) Limited attack diversity: The evaluation focuses on relatively narrow attack types (prompt injection and corpus poisoning at specific positions). More diverse or adaptive attack strategies—e.g., coordinated multi-position attacks or adversarial noise in high-ranked documents—are not explored in sufficient depth.

3) Heuristic design choices in sampling: Parameters in the sampling-based framework (e.g., decay rate γ, number of rounds T, context size m) are set heuristically. It would be valuable to either provide theoretical justification for these choices or to analyze their sensitivity more thoroughly.

4) Utility-robustness tradeoff under-explored: Although the authors attempt to mitigate benign utility loss (e.g., via “I don’t know” filtering), the risk of filtering relevant but noisy documents remains. The effect of such filtering in more ambiguous or low-resource scenarios is not fully addressed.

---

> ### Author Rebuttal · Authors · 2025-07-30
>
> We appreciate the reviewer for pointing out the potential weaknesses of our paper, and we will address these concerns by answering the questions.
>
> ---
>
> **Question 1**: How robust is the approach to NLI degradation? If the NLI model used for contradiction detection underperforms—e.g., on non-English or technical content—how significantly does this affect the reliability of the MIS defense?
>
> **Response to question 1:**
>
> We thank the reviewer for pointing out this potential issue. To address your concern, we have done the following additional experiments to demonstrate that the performance of our framework degrades gracefully with NLI degradation. In particular, we repeat our experiments on RealtimeQA and Mistral-7B with prompt injection attack, but impose some error probability on NLI checks. In particular, for each contradiction edge, we invert it with probability $\epsilon$ and test the performance of our framework under different values of $\epsilon$’s. The accuracy results of these additional experiments are as follows:
>
> With $k = 10$ retrieved documents:
>
> | $\epsilon$            | Attack @ Pos 1 | Attack @ Pos 5 | Attack @ Pos 10 |
> |--------------|---------------:|---------------:|----------------:|
> | 0.1          | 65             | 67             | 66              |
> | 0.3          | 64             | 62             | 62              |
> | 0.5          | 55             | 54             | 51              |
>
>
> With $k = 50$ retrieved documents:
>
> | $\epsilon$            | Attack @ Pos 1 | Attack @ Pos 25 | Attack @ Pos 50 |
> |--------------|---------------:|----------------:|----------------:|
> | 0.1          | 59             | 69              | 69              |
> | 0.3          | 62             | 67              | 65              |
> | 0.5          | 42             | 66              | 65              |
>
>
> We can see that, even with 0.5 error probability (i.e. we invert each edge with 50% probability), our methods still demonstrate superior robustness. Note that for $k = 50$, the accuracies are almost unaffected when the attack is on position 25 or 50 because the malicious document is very unlikely to get sampled to begin with. We can thus conclude that our methods are quite robust against NLI degradation.
>
> In addition, we would like to note that the possibility of NLI degradation is precisely why, on top of the empirical results, we have focused on providing theoretical analyses parameterized by the properties of the NLI performance. Furthermore, we have worked out and will provide new and tighter theoretical results and empirical simulations that augment this argument in later versions of our paper. Recognizing that malicious / low-quality documents might degrade the performance of NLI models, we provide a new version of Theorem 1 in which we permit larger error probability between benign documents and malicious documents compared with the current version of the paper. In particular, we show that even if the NLI model makes errors with ~0.3 probability between benign and malicious documents, our method is still $(1 - e^{O(k)})$-robust under certain natural conditions, where $k$ is the number of documents. We also conducted new simulations on the contradiction graph generation process (similar to Appendix B.1.1), and found that: even with ~0.4 NLI error probability between benign and malicious documents, the probability that any MIS includes any malicious document stays near zero until the number of malicious documents $k’$ becomes substantial relative to the total number of documents $k$. For example, robustness holds up to $k’ = 3$ for $k = 10$ and up to $k’ = 7$ for $k = 20$. Note that all of the above analysis assumes the worst case where any pair of malicious documents are non-contradictory to each other.
>
> Ultimately, we would like to mention that we are not relying on NLI model accuracy in isolation – our defence framework adopts the philosophy of defence-in-depth. The NLI signal is used in conjunction with the ranking signal, and both are inputs to the MIS algorithm;  so in order for adversaries to succeed, they need to exploit weaknesses in the NLI model and conduct adversarial search engine optimization, and do so in a way that subverts the protections provided by applying the MIS that uses both types of signals. Thus, although our approach works best when NLI is accurate, its defence-in-depth nature allows for gracefully handling of inaccuracies in NLI in the face of realistic adversaries.
>
> ---
>
> **Question 2**: Could the sampling parameters be learned or adapted? Is it feasible to tune the sampling distribution or context size in a data-driven manner, rather than fixing γ, T, and m heuristically?
>
> **Response to question 2:**
>
> We thank the reviewer for pointing out this question. This is a very promising direction for future work that generally falls into the learning-augmented algorithms paradigm, and we think is feasible. In our paper, we have also provided detailed ablation studies that could guide the choice of parameters in a data-driven and compute-aware manner:
>
> 1. For $\gamma$: In our experiments, we are using exponentially decaying weights with decay factor $\gamma$ as a proxy for reliability scores of documents. This is because we are using google retriever to retrieve relevant documents and that doesn’t come with an explicit reliability score. In real deployments of our system, we would expect retrieved documents to come with some reliability score / some meta-data that can offer a nice estimate for a reliability score, then one can just use these scores in place of the exponentially decaying weights in the paper.
>
> 2. For the number of sampling rounds $T$: As we have shown in Appendix C.6, the performance of our algorithm basically monotonically increases with $T$. In other words, increasing $T$ trades off compute for enhanced robustness. Thus, in real deployment, one can just decide on an appropriate value of $T$ by considering how much compute is available. One can find an appropriate $T$ with, say, binary search, to find the maximum acceptable $T$ with a given amount of compute. $T = 20$ turns out to be a good choice that nicely balances compute and performance in our experiments.
>
> 3. For the context size $m$: As we have shown in Appendix C.6, the performance generally decreases as $m$ increases, because a larger $m$ increases the likelihood of sampling malicious documents. Thus, if one expects there to be many potentially malicious / misleading documents, one might take a smaller $m$, e.g. $m = 2$ as in our experiments. If documents being malicious / misleading is unlikely, then one can take a larger $m$ as that avoids the algorithm from potentially missing relevant and useful documents.
>
> ---
>
> **Question 3**: Can the MIS selection be integrated with document reranking? For example, would combining MIS with re-ranking via learned relevance scores further enhance robustness or utility?
>
> **Response to question 3:**
>
> Yes, definitely. For example, if each document comes with a relevance / trustworthiness score, then we can construct a weighted contradiction graph where the weight of each document is its relevance / trustworthiness score. Then, we can find a weighted MIS instead of an unweighted one. Of course, the robustness / performance of this method depends on the quality of the relevance / trustworthiness score. If high-quality documents indeed have high scores, then the weighted MIS algorithm described above will more likely include them in the MIS, thus leading to enhanced robustness or utility. This is a very promising direction for future work, but out of the scope for the first paper in this space, as we wanted to demonstrate the promise of our method in the simplest conceptually possible scenario. Contradiction-based graph-theoretic methods should be a widely applicable technique that can see interesting applications in many different forms / domains.
>
> ---
>
> **Question 4**: What happens when no consistent majority exists? In cases where most retrieved documents are noisy, off-topic, or benignly contradictory, how does the system avoid excessive pruning or failure?
>
> **Response to question 4:**
>
> We appreciate the reviewer for pointing out this question, which points to a major motivation for why we would want to take reliability / rank information into account. We first note that completely noisy, off-topic documents are usually useless / could even have adverse effects for answering the question correctly and the best way to deal with them is to prune them out, regardless how our system works (e.g. [1] explicitly lists “context filtering” as a canonical sub‑module of RAG pipelines, alongside retrieval and generation).
>
> If no consistent majority exists among the relevant documents (after filtering out noisy, off-topic ones), then even humans may not effectively identify the correct answer from them. Our algorithm deals with this case gracefully though, as we explicitly prioritize higher-ranked documents. For example, say there are $k = 10$ documents and each document points to a different answer to the question. However, as long as the top-ranked document contains the correct answer, our algorithm can find it. In short, by explicitly utilizing rank / reliability information, our algorithm is still likely to find a decent answer even if a consistent majority is hasty.
>
> ---
>
> **References:**
>
> [1]. Sharma et al, Retrieval-Augmented Generation: A Comprehensive Survey of Architectures, Enhancements, and Robustness Frontiers

---

> > ### Comment · Reviewer_Xjz6 · 2025-08-09
> >
> > Thank you for the detailed clarifications provided in response to my questions. The explanations have helped address several of my concerns. However, I noticed that the authors did not address the limitation I previously pointed out. If the authors could provide a further clarification or response on this point, I would be open to reconsidering and potentially adjusting my score.

---

> > > ### Author Response · Authors · 2025-08-09
> > >
> > > We appreciate the reviewer for the followup, and we apologize for having not responded to the limitations in addition to the questions, which are all great questions.
> > >
> > > ---
> > >
> > > **Limitation 1:** The approach presumes the existence of a coherent, contradiction-free majority of retrieved documents ...
> > >
> > > **Response to limitation 1:**
> > >
> > > This is a great question. In the real world, queries can be ambiguous or multi-perspective. One can potentially build a more flexible framework off of our MIS approach. For example, one can detect whether several MIS exist with comparable rank / weight and either (i) provide a multi-view answer that summarizes each consistent cluster or (ii) ask a clarifying question when clusters are highly incompatible. This is one of many ways one can employ to build a reliable agentic system using our MIS framework. One can also replace the hard NLI thresholds ($\beta = 0.5$) with a weighted graph where edge weights come from NLI scores and select a maximum-weight, low-cut subgraph. However, these might be out of the scope of the current paper, and the question of how to extend our MIS framework to more complex scenarios is a very interesting and fruitful direction of future research. Our work offers both theoretical and practical grounding for this framework, and we expect to see broader applications of this simple and interpretable approach in more complex scenarios in the future.
> > >
> > > ---
> > >
> > > **Limitation 2:** The scalability of the full MIS procedure is limited to relatively small k (e.g., ≤ 20) ...
> > >
> > > **Response to limitation 2:**
> > >
> > > As we have discussed in the response to your question 2, tuning the sampling distribution or context size in a data-driven manner is a very promising direction for future work that generally falls into the learning-augmented algorithms paradigm and we believe is feasible. In addition, in our paper, we have also provided detailed ablation studies that could guide the choice of parameters in a data-driven and compute-aware manner.
> > >
> > > Ultimately, the weighted sampling framework in the form we present is just one way to scale up MIS, and we choose to present it because it is both provably robust (as shown in Appendix B) and works well in practice. There are many potential ways to scale up the MIS framework: For example, one can use effective heuristics such as LP rounding or Luby’s algorithm to compute an approximate MIS. One can also use sampling to implement a natural iterative filtering process: In each iteration, we sample, say, 20 or 30 documents (the maximum size of a graph on which we can compute an exact MIS) and filter out the non-MIS documents. All these are interesting and nice heuristics that we believe can effectively scale up MIS and are fruitful directions for future research, but might be out of the scope of the current paper. That said, we believe the MIS framework is very valuable both for its superior performance and for its extensibility, as it has the potential to become the foundation of a new class of robust defence algorithms.
> > >
> > > ---
> > >
> > > **Limitation 3:** The reliance on search engine ranking as a proxy ...
> > >
> > > **Response to limitation 3:**
> > >
> > > For settings outside of web search, such as academia corpora, social media, or enterprise knowledge base, the retrieved documents usually come with meta data that specifies their source, publisher, authors & affiliations, published date, document type, license, number of citations, etc, many of which are great sources for estimating the reliability of the retrieved document.
> > >
> > > ---
> > >
> > > **Limitation 4:** The filtering mechanism (based on “I don’t know” responses) ...
> > >
> > > **Response to limitation 4:**
> > >
> > > One can definitely replace the “I don’t know” filter with non-LLM gates: For example, one can use a calibrated reranker score threshold or a lightweight relevance / quality classifier (e.g., CRAG-style retrieval evaluator [1]). In addition, there is a solid body of works in information retrieval literature that show that LLMs can effectively judge (and thus filter) non-relevant documents well, such as [2], [3], and [4]. We also note that it is not necessary to run this pre-filtering step with exactly the same LLM as the generation LLM; one can definitely fine-tune a specific model for this particular step for consistent and high-quality filtering.
> > >
> > > ---
> > >
> > > We thank the reviewer for pointing out these potential limitations, and we will better address these points in later versions of our paper.
> > >
> > > ---
> > >
> > > **References:**
> > >
> > > [1]. Yan el al, Corrective Retrieval Augmented Generation
> > >
> > > [2]. Nouriinanloo et al, Re-Ranking Step by Step: Investigating Pre-Filtering for Re-Ranking with Large Language Models
> > >
> > > [3]. Upadhyay et al, A Large-Scale Study of Relevance Assessments with Large Language Models: An Initial Look
> > >
> > > [4]. Sun et al, Is ChatGPT Good at Search? Investigating Large Language Models as Re-Ranking Agents

---

> ### Comment · Area_Chair_s71Z · 2025-08-08
> **AC Message**
>
> Dear Reviewer Xjz6,
>
> The authors have already submitted their response to your review, and we are approaching the deadline for author-reviewer discussions (August 8, 11:59 PM AoE).
>
> Could you please share your feedback with the authors regarding whether their response sufficiently addresses the concerns raised in your review, and if any issues remain that require further clarification or elaboration?
>
> As a key reminder, active engagement in the discussion process is critical at this stage. Reviewers must participate in discussions with authors (e.g., by posting follow-up feedback in the discussion thread), and then submit a Mandatory Acknowledgement to confirm your participation.
>
> Best regards,
>
> AC

---

### Official Review · Reviewer_pm3b · 2025-07-01

**Clarity:** 4
**Significance:** 4
**Originality:** 3
**Rating:** 4
**Confidence:** 4

**Summary:**

This work introduces a novel defense against RAG retrieval attacks by leveraging document reliability signals (rank/weights) through two key mechanisms: (1) Maximum Independent Set (MIS) algorithm that selects mutually consistent documents while prioritizing high-reliability sources, and (2) Scalable weighted sampling framework for large retrieval sets.
Experiments demonstrate good robustness against prompt injection/corpus poisoning attacks while maintaining high benign performance, especially in long-form generation.

**Questions:**

Can you measure the time overhead of actually using the framework and whether it will significantly affect the user experience?

**Ethical Concerns:**

["NO or VERY MINOR ethics concerns only"]

**Final Justification:**

Thanks to the author for the detailed rebuttal. It can solve my concerns to some extent. The paper could be improved by considering more application scenarios.

**Limitations:**

Although the framework looks effective, it lacks an explanation of the actual application capabilities, and adding this would add real meaning to the work.

**Quality:**

3

**Strengths And Weaknesses:**

Strengths:
1. First framework to exploit document ranks/weights for provable robustness against retrieval attacks.The idea is simple and straightforward but seems to work;
2. Weighted sampling extends MIS robustness to large corpora with minimal accuracy loss;
3. Maintain good robustness against attacks while excelling in long-form generation.
Weakness:
1.  Robustness hinges on NLI model accuracy; vulnerable to adversarial contradictions;
2.  MIS and NLI checks add overhead , limiting real-time use;
3.  No validation for scenarios where reliability signals (ranks/weights) are miscalibrated or manipulated.

---

> ### Author Rebuttal · Authors · 2025-07-30
>
> **Weakness 1**: Robustness hinges on NLI model accuracy; vulnerable to adversarial contradictions.
>
> **Response to weakness 1:**
>
> We thank the reviewer for pointing out this potential issue. To address your concern, we have done the following additional experiments to demonstrate that the performance of our framework degrades gracefully with NLI degradation. In particular, we repeat our experiments on RealtimeQA and Mistral-7B with prompt injection attack, but impose some error probability on NLI checks. In particular, for each contradiction edge, we invert it with probability $\epsilon$ and test the performance of our framework under different values of $\epsilon$’s. The accuracy results of these additional experiments are as follows:
>
> With $k = 10$ retrieved documents:
>
> | $\epsilon$            | Attack @ Pos 1 | Attack @ Pos 5 | Attack @ Pos 10 |
> |--------------|---------------:|---------------:|----------------:|
> | 0.1          | 65             | 67             | 66              |
> | 0.3          | 64             | 62             | 62              |
> | 0.5          | 55             | 54             | 51              |
>
> With $k = 50$ retrieved documents:
>
> | $\epsilon$            | Attack @ Pos 1 | Attack @ Pos 25 | Attack @ Pos 50 |
> |--------------|---------------:|----------------:|----------------:|
> | 0.1          | 59             | 69              | 69              |
> | 0.3          | 62             | 67              | 65              |
> | 0.5          | 42             | 66              | 65              |
>
> We can see that, even with 0.5 error probability (i.e. we invert each edge with 50% probability), our methods still demonstrate superior robustness. Note that for $k = 50$, the accuracies are almost unaffected when the attack is on position 25 or 50 because those documents are very unlikely to get sampled to begin with. We can thus conclude that our methods are quite robust against NLI degradation.
>
> In addition, we would like to note that the possibility of NLI degradation is precisely why, on top of the empirical results, we have focused on providing theoretical analyses parameterized by the properties of the NLI performance. Furthermore, we have worked out and will provide new and tighter theoretical results and empirical simulations that augment this argument in later versions of our paper. Recognizing that malicious / low-quality documents might degrade the performance of NLI models, we provide a new version of Theorem 1 in which we permit larger error probability between benign documents and malicious documents compared with the current version of the paper. In particular, we show that even if the NLI model makes errors with ~0.3 probability between benign and malicious documents, our method is still $(1 - e^{O(k)})$-robust under certain natural conditions, where $k$ is the number of documents. We also conducted new simulations on the contradiction graph generation process (similar to Appendix B.1.1), and found that: even with ~0.4 NLI error probability between benign and malicious documents, the probability that any MIS includes any malicious document stays near zero until the number of malicious documents $k’$ becomes substantial relative to the total number of documents $k$. For example, robustness holds up to $k’ = 3$ for $k = 10$ and up to $k’ = 7$ for $k = 20$. Note that all of the above analysis assumes the worst case where any pair of malicious documents are non-contradictory to each other.
>
> Ultimately, we would like to mention that we are not relying on NLI model accuracy in isolation – our defence framework adopts the philosophy of defence-in-depth. The NLI signal is used in conjunction with the ranking signal, and both are inputs to the MIS algorithm;  so in order for adversaries to succeed, they need to exploit weaknesses in the NLI model and conduct adversarial search engine optimization, and do so in a way that subverts the protections provided by applying the MIS that uses both types of signals. Thus, although our approach works best when NLI is accurate, its defence-in-depth nature allows for gracefully handling of inaccuracies in NLI in the face of realistic adversaries.
>
> ---
>
> **Weakness 2**: MIS and NLI checks add overhead , limiting real-time use.
>
> **Response to weakness 2:**
>
> We thank the reviewer for pointing out this potential issue, and we will provide more detailed discussions in the response to your Question 1. For here, we would like to clarify that we did provide detailed discussions of the overhead of our framework in Appendix D.2. In particular, even with our current setup (which is not very optimized in terms of running time), NLI checks induce nearly negligible overhead (< 0.05s for even $k = 50$ retrieved documents). The “isolated answering” stage and MIS construction contributes to a major part of the overhead, but is still far from limiting real-time use (only around 0.2 ~ 0.4s).
>
> ---
>
> **Weakness 3**: No validation for scenarios where reliability signals (ranks/weights) are miscalibrated or manipulated.
>
> **Response to weakness 3:**
>
> We thank the reviewer for pointing out this potential issue. We first note that these scenarios are somewhat out of the scope of our current work, because one major novelty of our methods is that the ranks are difficult to compromise in the search context (e.g. in google retrievers).
>
> In addition, we would like to note that our methods could offer decent solutions even in the case where reliability signals are compromised.
>
> In MIS, we are essentially taking a majority vote over the documents, so even if the reliability signal of a few documents are manipulated, as long as benign ones still form a majority, our algorithm will return the correct answer. If the benign ones are not the majority to begin with and, furthermore, malicious documents manage to climb up in the rank, then even humans can find it hard to effectively identify the correct answer.
>
> In Sampling + MIS, one might control the decay factor $\gamma$ to decide how much to trust higher-ranked documents compared to lower-ranked documents. In scenarios where reliability signals can be easily compromised or are known to be of low quality, one might place more uniform weights across the documents to avoid overly trusting the higher-ranked ones.
>
> ---
>
> **Question 1**: Can you measure the time overhead of actually using the framework and whether it will significantly affect the user experience?
>
> **Response to question 1:**
>
> We would like to clarify that we did provide a detailed discussion of overhead issues and potential strategies to mitigate this in Appendix D.2. In that section, we show that the running time of our entire framework is ~0.61s for $k = 10$ retrieved documents + Mistral-7B, ~0.41s for $k = 10$ retrieved documents and Llama-3B, ~1.32s for $k = 50$ retrieved documents + Mistral-7B, and ~0.92s for $k = 50$ retrieved documents + Llama-3B, with vLLM and simple parallelism like batch query. Moreover, this is just based on our prototype academic setup and can be, of course, accelerated even further with more proper parallelization and tighter system integration.
>
> To potentially reduce the latency overhead even more, one can perform the “isolated answering” stage using a smaller, faster language model instead of the LLM for the RAG query. Such a model could rapidly assess documents for basic contradictions or irrelevance. This is likely sufficient for detecting rudimentary issues such as simple prompt injections or factual poisoning, but more targeted and nuanced attacks may bypass the filter, requiring careful consideration based on the specific threat model and application context.
>
> Even with the current latency with our preliminary setup which induces a sub-1s latency, it will not significantly affect the user experience. Users interacting with sophisticated RAG systems often experience multi-second or even minute latencies, which can stem not only from retrieval and basic generation but also from extensive downstream processing, such as reasoning or other test-time scaling techniques applied for enhanced analysis and answer quality. Again, we would like to note that our current setup is not very optimized in terms of time overhead, as the major goal of our paper is to show the value of our contradiction-based graph-theoretic approach.

---

> > ### Comment · Reviewer_pm3b · 2025-08-08
> >
> > Thank you, and apologies for the delayed response. I truly appreciate the high-quality rebuttal. The rebuttal have addressed my concerns to some extend. I will keep my score.

---

### Official Review · Reviewer_ei33 · 2025-07-02

**Clarity:** 3
**Significance:** 3
**Originality:** 3
**Rating:** 5
**Confidence:** 3

**Summary:**

The authors propose a defence framework for RAG systems that explicitly exploits document‐level reliability signals to mitigate corpus-level attacks like prompt injection and poisoning. Their core idea is to identify a consistent majority of mutually non-contradictory, high-reliability documents before passing them to the language model. To do so, they construct a contradiction graph over the retrieved passages using a NLI model and then select a rank-aware maximum independent set (MIS); when several MIS of equal size exist, the algorithm chooses the lexicographically highest-ranked one, thereby prioritising trustworthy sources.

**Questions:**

You make no assumption when both documents are malicious. In practice, how often did your attacks produce malicious pairs that the NLI judged “non-contradictory,” and does this undermine robustness?

Did you perform sensitivity analysis on beta, and is there a principled way to adapt it per query or domain?

More broadly for parameters please clarify choices made for beta as mentioned, m, t etc.

**Ethical Concerns:**

["NO or VERY MINOR ethics concerns only"]

**Final Justification:**

From author clarifications and verification through additional experiments, I see the paper in a more positive light. Some concerns are more difficult to address than others as the issue may not be with the paper itself but in the paper following current practise which may be unsound.

**Limitations:**

Yes

**Quality:**

3

**Strengths And Weaknesses:**

Strengths
- Clear motivation and well-structured investigation
- The work provides a clear framework for robustness checks exploiting previously ignored signals
- The addition of another NLI model does indeed improve robustness while naturally improving effectiveness

Weaknesses
- At least 2 decimal points and CI / Significance would be helpful in presenting convincing evidence in the main results, generally tests have been applied sporadically.
- The motivation (L 62-63) that higher results are more trustworthy appears to conflate cognitive bias and causal factors, while this may hold due to popularity signals in a commercial setting I feel this motivation should be clarified in an ad-hoc setting. Generally the work is well-specified in scope so this is a nitpick but may be helpful.
- The approach could be contrasted with more naive signals, given the motivation in a commercial setting why not apply naive lexical matching and observe if any attack can even succeed?
- The lack of human annotation (solely llm-as-a-judge) prevents evidence from being particularly compelling, simply using an openQA dataset which uses for example multiple-choice answers such that llm-as-a-judge is unneeded would be sufficient to strengthen evidence.
- The latency experiments without explicit weakness discussion are not convincing but with clearer discussion and acknowledgement that 4x RAG is not satisfactory in most settings, this may be considered a minor point.

Note
- The use of the NLI model appears somewhat similar to broader static rank, it would be interesting to observe comparisons with these models to consider if a neural model (and thus massive overhead in pair-wise comparisons) is needed.

---

> ### Author Rebuttal · Authors · 2025-07-30
>
> **Weakness 1**: At least 2 decimal points and CI / Significance would be helpful in presenting convincing evidence in the main results, generally tests have been applied sporadically.
>
> **Response to weakness 1:**
>
> We thank the reviewer for the suggestion, and we apologize that we were not able to repeat the experiments for multiple times for all experiments in our paper due to budget constraints (as we were using GPT-4o-as-a-judge to judge correctness of answers for high-quality judgement, which was quite expensive for large-scale experiments). We note that we did provide results with error bars in Appendix C.4 for $k = 50$ documents, Mistral-7B on all datasets. As presented in Figure 4 and 5, the error bars are very narrow, and our method consistently performs better than all the other baselines. We appreciate the reviewer for pointing this out, and we will provide more discussions about the statistical significance of our results.
>
> ---
>
> **Weakness 2**: The motivation (L 62-63)...
>
> **Response to weakness 2:**
>
> We thank the reviewer for pointing this out, and we will revise Line 62-63 to make this distinction clear.
>
>
> ---
>
> **Weakness 3**: The approach could be contrasted with more naive signals, given the motivation in a commercial setting why not apply naive lexical matching and observe if any attack can even succeed?
>
> **Response to weakness 3:**
>
> We thank the reviewer for raising this question. We have two interpretations for what the reviewer might have had in mind by contrasting with applying naive lexical matching. One interpretation is that it means checking how well each document lexically matches the prompt and selecting the document that matches the most. Such approaches are known to be vulnerable to adversarial attacks,  and prior works show attacks with documents that are lexically / semantically close to the prompt (e.g., [3]); thus lexical matching may be a useful, but not sufficient signal. Our second interpretation is that the reviewer suggests replacing the NLI with using lexical matching to detect contradictions / non-contradictions. This is, of course, possible, and can be viewed as a simplified version of our approach. We are not sure whether we misinterpreted the suggestion, and we would appreciate it if the reviewer could clarify the approach they suggest they suggest we contrast with so that we can provide additional experimental results for it.
>
> ---
>
> **Weakness 4**: The lack of human annotation (solely llm-as-a-judge) prevents evidence from being particularly compelling, simply using an openQA dataset which uses for example multiple-choice answers such that llm-as-a-judge is unneeded would be sufficient to strengthen evidence.
>
> **Response to weakness 4:**
>
> We thank the reviewer for pointing out this potential complication. We first note that llm-as-a-judge and limited datasets are a function of the cost of running such experiments, and a commercial lab less limited by costs could do it. In addition, using llm-as-a-judge (in particular, GPT-4o-as-a-judge) is already much more rigorous than many existing works with QA-dataset evaluations, such as [1] which uses direct string comparisons. In addition, using llm-as-a-judge is a widely adopted method for judging correctness of answers for QA datasets, e.g. by OpenAI ([2]). Furthermore, we human-evaluated 100 llm-judged answers on the RealtimeQA dataset, and found that all the judgments are correct.
>
> Finally, to better address your concern, we also conduct additional experiments by constructing a multiple-choice dataset using our RealtimeQA dataset and test the performance of our methods compared against baseline methods (with Mistral-7B and prompt injection attack, using the same setup of experiments as presented in our paper):
>
> With $k = 10$ retrieved documents:
>
> | Method        | Attack @ Pos 1 | Attack @ Pos 5 | Attack @ Pos 10 |
> |-------------------------|---------------:|---------------:|----------------:|
> | MIS                     | 65             | 70             | 68              |
> | RobustRAG (keyword)     | 62             | 65             | 50              |
> | VanillaRAG              | 51             | 51             | 19              |
> | InstructRAG             | 64             | 56             | 27              |
> | AstuteRAG               | 30             | 22             | 16              |
>
> With $k = 50$ retrieved documents:
>
> | Method       | Attack @ Pos 1 | Attack @ Pos 25 | Attack @ Pos 50 |
> |-------------------------|---------------:|----------------:|----------------:|
> | SampleMIS               | 70             | 77              | 79              |
> | RobustRAG (keyword)     | 55             | 67              | 63              |
> | VanillaRAG              | 54             | 45              | 16              |
> | InstructRAG             | 58             | 42              | 16              |
> | AstuteRAG               | 29             | 17              | 16              |
>
> We can see that our methods perform similarly well compared with other baseline methods in this multiple-choice setting, which offers additional evidence for the strength of our methods. We appreciate the reviewer for pointing out the value of showing this, and we will add these results and relevant discussions into later versions of our paper.
>
> ---
>
> **Weakness 5**: The latency experiments without explicit weakness discussion are not convincing but with clearer discussion and acknowledgement that 4x RAG is not satisfactory in most settings, this may be considered a minor point.
>
> **Response to weakness 5:**
>
> We acknowledge that our method is inducing additional latency, but we would also like to clarify that this is just based on our prototype setup and can be accelerated more with proper parallelization and tighter system integration. To potentially reduce the latency overhead even more, one can perform the “isolated answering” stage using a smaller, faster language model instead of the LLM for the RAG query.
>
> We note that even with the current latency of our preliminary setup which induces a sub-1s latency, it will not significantly affect the user experience. Users interacting with sophisticated RAG systems often experience multi-second or even minute latencies, which can stem not only from retrieval and basic generation but also from extensive downstream processing, such as reasoning or other test-time scaling techniques applied for enhanced analysis and answer quality.
>
> ---
>
> **Response to the Note:**
>
> We appreciate the reviewer for pointing out this relation, which is an interesting direction of future work. We would also like to clarify that, as presented in Appendix D.2, the pairwise NLI checks induce nearly negligible (instead of massive) overhead (< 0.05s for even $k = 50$ retrieved documents)
>
> ---
>
> **Question 1**: You make no assumption when both documents are malicious. In practice, how often did your attacks produce malicious pairs that the NLI judged “non-contradictory,” and does this undermine robustness?
>
> **Response to question 1:**
>
> We clarify that in our theoretical analysis (Theorem 1), we assume the worst-case scenario where all malicious pairs are judged as non-contradictory, as adversarial behaviors can induce outcomes that can be arbitrarily bad. Even under this pessimistic assumption, our method shows nice theoretical guarantees. In reality, (1) In single-position attacks, there’s only one malicious document so there are no malicious pairs here. (2) In multi-position attacks, all malicious documents are constructed in such a way that they point to the same malicious answer, so any pair of malicious documents are non-contradictory, i.e., we consider the worst-case scenario in our design of experiments as well. As presented in Appendix C.5, our methods show excellent robustness against multi-position attacks, even when all pairs of malicious documents are non-contradictory. To summarize, the potential issue of malicious pairs being non-contradictory is studied and addressed both theoretically and experimentally in our paper. We apologize if our current presentation is unclear regarding this point, and we will make this clearer in later versions of our paper.
>
> ---
>
> **Question 2**: Did you perform sensitivity analysis on beta, and is there a principled way to adapt it per query or domain?
>
> **Response to question 2:**
>
> Yes, we did experiment with different values of $\beta$, from 0.2 to 0.8. One observation is that the output of the NLI model (contradiction probability) is usually almost binary (very close to 0 or 1), so our results are not very sensitive to the choice of $\beta$. Since 0.5 is a natural and previously adopted choice (e.g., in [1]), we present $\beta = 0.5$ in the paper. As per our observation in experiments (which ranges many different domains and queries), there is no need for specific adaptation per query or domain. We thank the reviewer for pointing out the value of sensitivity analysis here, and we will add them into later versions of our paper.
>
> ---
>
> **Question 3**: More broadly for parameters please clarify choices made for beta as mentioned, m, t etc.
>
> **Response to question 3:**
>
> We appreciate the reviewer for pointing out the importance of clarifying design choices made, and we will add more discussions regarding them. We presented extensive ablation studies regarding the choice of parameters in Appendix C.6 and discussions of their impacts, including (1). The impact of varying context size $m$; (2). The impact of varying the number of sampling rounds $T$; (3). The impact of varying the decaying factor $\gamma$; (4). The impact of weight decaying schemes.
>
> ---
>
> References:
>
> [1] Xiang et al, Certifiably Robust RAG against Retrieval Corruption
>
> [2] OpenAI, openai/simple-evals Github Repository
>
> [3] Zou et al, PoisonedRAG: Knowledge Corruption Attacks to Retrieval-Augmented Generation of Large Language Models

---

> > ### Comment · Reviewer_ei33 · 2025-08-03
> > **Response to Rebuttal**
> >
> > My thanks to the authors for their thorough response. With respect to weakness 3, I rather like the alternate interpretation chosen by the authors. However, what I meant was, often in an adversarial setting in ranking, we ignore that many larger systems are built first to retrieve k documents using term matching heuristics to then be re-ranked. Depending on the adversarial process, this term matching could be harmed. Thus, I would be interested in seeing how lexical match (say, under Porter stemming) is affected by attacks. I am not asking for a full re-evaluation with a different pipeline, but would instead ask that stemming is applied to contrast the attacked text with the original in terms of their query match.
> >
> > Overall, the response is thorough, and I will raise my score. Conditional on the above discussion and subsequent possible experiments, I could raise it further.

---

> > > ### Author Response · Authors · 2025-08-04
> > >
> > > We appreciate your clarification, which is very helpful for us. We have conducted additional experiments to evaluate whether a lexical filter built on Porter stemming would work well to counter our attacks, and we find that it does not. For each text corpus (e.g. a query or a retrieved document), we use the canonical `nltk.stem.PorterStemmer` implementation to compute its stem set. Then, for each query-document pair, we compute the Jaccard overlap $J(Q, P) = \frac{|Q \cap P|}{|Q \cup P|}$ between the stem set of the query and the stem set of the retrieved document. We choose the Jaccard overlap between two sets since it is  a common practice in information-retrieval for quantifying lexical similarity [1]. Given a query with stem set $Q$, for a document with stem set $P$ and its poisoned counterpart with stem set $P’$, we declare that **the attack fails the lexical check if $J(P, Q) > \delta$ while $J(P’, Q) < \delta$ for some chosen threshold $\delta$**.
> > >
> > > For each $\delta \in$ {0.1, 0.2, ..., 0.9}, on RealtimeQA dataset (with 100 queries and 4734 retrieved documents in total), the attack fails the lexical check at the following rates:
> > >
> > > With prompt injection attack:
> > >
> > > | $\delta$ | Failure&nbsp;rate&nbsp;(%) | Number&nbsp;of&nbsp;attacks&nbsp;that&nbsp;fail&nbsp;check | Total |
> > > |---------:|-------------------------:|-------------------------------------:|------:|
> > > | 0.1 | 0.00 | 0 | 4734 |
> > > | 0.2 | 0.00 | 0 | 4734 |
> > > | 0.3 | 0.00 | 0 | 4734 |
> > > | 0.4 | 0.02 | 1 | 4734 |
> > > | 0.5 | 0.21 | 10 | 4734 |
> > > | 0.6 | 0.30 | 14 | 4734 |
> > > | 0.7 | 0.32 | 15 | 4734 |
> > > | 0.8 | 0.21 | 10 | 4734 |
> > > | 0.9 | 0.11 | 5 | 4734 |
> > >
> > >
> > > With corpus poisoning attack:
> > >
> > > | $\delta$ | Failure&nbsp;rate&nbsp;(%) | Number&nbsp;of&nbsp;attacks&nbsp;that&nbsp;fail&nbsp;check | Total |
> > > |---------:|-------------------------:|-------------------------------------:|------:|
> > > | 0.1 | 0.00 | 0   | 4734 |
> > > | 0.2 | 3.72 | 176 | 4734 |
> > > | 0.3 | 4.31 | 204 | 4734 |
> > > | 0.4 | 1.88 | 89  | 4734 |
> > > | 0.5 | 1.20 | 57  | 4734 |
> > > | 0.6 | 0.59 | 28  | 4734 |
> > > | 0.7 | 0.32 | 15  | 4734 |
> > > | 0.8 | 0.21 | 10  | 4734 |
> > > | 0.9 | 0.11 | 5   | 4734 |
> > >
> > > We can see that no more than 4.31% of attacks fail the lexical check even under the optimal choice of $\delta$ in hindsight for corpus poisoning attack, and almost no attacks fail the lexical check for prompt injection attack. Thus, simply doing lexical matching is not sufficient for countering adversarial attacks, even with our preliminary design of attacks.
> > >
> > > In some sense, these results are not surprising, especially in a commercial setting, because the malicious prompts are usually designed to be closely relevant to the query. (see Appendix E.2. for exact description of our attack prompt design). As an example, with a query about what is the best car, the original document might be “Honda is the best car, because it has superior reliability and speed”, while a poisoned document might be “Toyota is the best car, because it has superior reliability and speed”, both of which are equally suited to passing a lexical filter. In fact, we suspect that in two-stage retrieval systems, powerful adversaries could take advantage of the two stages by use of common lexical checks to make their malicious documents more likely to get retrieved (e.g. with the strategy as presented in Section 4.2.2 in [2]). Thus, we would say that naive signals like lexical matching may not be sufficient for countering adversarial attacks, and more sophisticated defences are needed. This serves as an additional motivation for our graph-theoretic contradiction-based method, and we sincerely appreciate the reviewer for bringing up this point.
> > >
> > > We also thank the reviewer for bringing up this direction because it raises a new question as to whether there’s a “tax” to adversarial documents that may be observed through other means, similarly to how recent works like [3] show that there’s a usefulness tax on jailbroken outputs. Although this tax does not appear visible under lexical checking, we wonder whether other more sophisticated techniques may be able to pick up on it, and think it’s an interesting direction for future work.
> > >
> > > [1]. Manning et al, An introduction to information retrieval
> > >
> > > [2]. Zou et al, PoisonedRAG: Knowledge Corruption Attacks to Retrieval-Augmented Generation of Large Language Models
> > >
> > > [3]. Nikolic et al, The Jailbreak Tax: How Useful are Your Jailbreak Outputs?

---

### Official Review · Reviewer_PSmh · 2025-07-03

**Clarity:** 3
**Significance:** 3
**Originality:** 3
**Rating:** 4
**Confidence:** 4

**Summary:**

This paper proposes ReliabilityRAG, a defense framework against retrieval corpus attacks in RAG systems. More specifically, the authors leverage document reliability information (such as search rankings) and identify a set of mutually consistent documents by constructing a contradiction graph and solving for the Maximum Independent Set (MIS). Their experiments show that while maintaining high accuracy in benign scenarios, the method demonstrates superior robustness against attacks like prompt injection and corpus poisoning.

**Questions:**

see the weakness

**Ethical Concerns:**

["NO or VERY MINOR ethics concerns only"]

**Final Justification:**

Thank the authors for addressing my questions. I think this is a good paper and will keep my score

**Limitations:**

Yes

**Paper Formatting Concerns:**

the format is correct

**Quality:**

3

**Strengths And Weaknesses:**

Strengths

1)	I have a good impression of this paper - adversarial defense for RAG systems is an important research topic.

2)	Transforming the document consistency problem into a maximum independent set problem on graphs is a very clever idea. The methodology of constructing contradiction graphs based on NLI and applying MIS is both solid and practically actionable.

3)	Nice combination of theory and practice: The paper not only provides theoretical guarantees of λ-robustness (Theorem 1), but also addresses computational efficiency for large document sets through the weighted sample and aggregate framework, showing good practicality.

Weaknesses

1)	Did the authors try applying different NLI models (I'm not sure if there are more NLI options available), and did they do sensitivity analysis on the contradiction threshold β? I'd like to see more ablation results.

2)	The analysis of multi-position attacks could be more in-depth. Section 5.4 only shows partial results - I suggest providing more analysis in the main text about system behavior when attackers simultaneously corrupt documents at multiple different ranking positions. This would help understand the limits of the defense mechanism.

3)	Some detail issues:
	The isolated answering step in Algorithm 1 incurs additional LLM call overhead - can this be reduced through caching or other optimizations? Some discussion and insights would be helpful.

	In Figure 2's caption, "attacked documents" should clearly specify that it's a continuous suffix attack pattern.

	From a paper formatting perspective, I suggest the authors not use paragraph breaks in the abstract and make it more concise (the current abstract is too long).

	Similarly, I hope the authors carefully polish the introduction section. While I understand that using bold text and hard paragraph breaks can more clearly express related work, for a research paper, I'd prefer the introduction to have better overall coherence and narrative flow.

---

> ### Author Rebuttal · Authors · 2025-07-30
>
> **Weakness 1**: Did the authors try applying different NLI models (I'm not sure if there are more NLI options available), and did they do sensitivity analysis on the contradiction threshold β? I'd like to see more ablation results.
>
> **Response to weakness 1:**
>
> Did the authors try applying different NLI models - Yes, we did experiment with other NLI models like deberta-v3-large-mnli, and that yields similarly good results. In particular, on RealtimeQA, with Mistral-7B and under prompt injection attack, the accuracy results for our techniques are as follows:
>
> | Setting ($k = 10$) | Attack @ Pos 1 | Attack @ Pos 5 | Attack @ Pos 10 |
> |------------------|---------------:|---------------:|----------------:|
> | MIS              | 68             | 68             | 62              |
>
> | Setting ($k = 50$) | Attack @ Pos 1 | Attack @ Pos 25 | Attack @ Pos 50 |
> |------------------|---------------:|----------------:|----------------:|
> | Sampling + MIS   | 53             | 70              | 71              |
>
> In general, we expect (and our theoretical results support this) most NLI models with decent performance to yield similarly good results.
>
> Did they do sensitivity analysis on $\beta$ - Yes, we did experiment with different values of $\beta$, from 0.2 to 0.8. One observation is that the output of the NLI model (contradiction probability) is usually almost binary (very close to 0 or 1), so our results are not very sensitive to the choice of $\beta$. Since 0.5 is a natural and previously adopted choice (e.g., in [1]), we present $\beta = 0.5$ in the paper.
>
> We thank the reviewer for pointing out the value of more ablation studies, and we will add them into later versions of our paper.
>
> ---
>
> **Weakness 2**: The analysis of multi-position attacks could be more in-depth. Section 5.4 only shows partial results - I suggest providing more analysis in the main text about system behavior when attackers simultaneously corrupt documents at multiple different ranking positions. This would help understand the limits of the defense mechanism.
>
> **Response to weakness 2:**
>
> We appreciate this suggestion, and we sincerely apologize that we were not able to put more results about multi-position attacks in the main body of the paper due to limited space. More comprehensive analysis and discussions are presented in Appendix C.5, where we experimented with different corruption sizes (0, 1, 2, 3, 4 for $k = 10$ retrieved documents and 0, 5, 10, 15, 20 for $k = 50$ retrieved documents) on all the datasets. Our MIS / Sampling + MIS almost consistently perform significantly better than RobustRAG (Keyword), and show excellent robustness even when almost half of the documents are malicious.
>
> We thank the reviewer for pointing out the importance of this part, and we will add more discussions about multi-position attacks in the main body in later versions of our paper.
>
> ---
>
> **Weakness 3.1**: Some detail issues: The isolated answering step in Algorithm 1 incurs additional LLM call overhead - can this be reduced through caching or other optimizations?
>
> **Response to weakness 3.1:**
>
> Can overhead be reduced through caching or other optimizations - Yes, definitely. We did provide a detailed discussion of overhead issues and potential strategies to mitigate this in Appendix D.2. In that section, we show that the overhead of the “isolated answering” step is ~0.27s for $k = 10$ retrieved documents + Mistral-7B, ~0.16s for $k = 10$ retrieved documents and Llama-3B, ~0.38s for $k = 50$ retrieved documents + Mistral-7B, and ~0.20s for $k = 50$ retrieved documents + Llama-3B, with vLLM and simple parallelism like batch query. Moreover, this is just based on our prototype setup and can be accelerated even more with more proper parallelization and tighter system integration.
>
> ---
>
> **Weakness 3.2**: Some discussion and insights would be helpful. In Figure 2's caption, "attacked documents" should clearly specify that it's a continuous suffix attack pattern.
>
> **Response to weakness 3.2:**
>
> We thank the reviewer for the suggestion, and we will change Figure 2’s caption to make it clear that we are considering a continuous suffix attack. We sincerely apologize that we were not able to provide more discussion and insights in the main body of the paper due to limited space;  we placed more discussions (including more experiments, extensive ablation studies, and adaptive attack) in Appendix C. We will add more discussions and insights in later versions of our paper and in the main text.
>
> ---
>
> **Weakness 3.3**: From a paper formatting perspective, I suggest the authors not use paragraph breaks in the abstract and make it more concise (the current abstract is too long).
>
> **Response to weakness 3.3:**
>
> We thank the reviewer for the suggestion, and we will prune and reformat the abstract in later versions of our paper.
>
> ---
>
> **Weakness 3.4**: Similarly, I hope the authors carefully polish the introduction section. While I understand that using bold text and hard paragraph breaks can more clearly express related work, for a research paper, I'd prefer the introduction to have better overall coherence and narrative flow.
>
> **Response to weakness 3.4:**
>
> We thank the reviewer for the suggestion. We will refine the introduction section in later versions of our paper to make it more coherent and narrative.
>
> ---
>
> **References:**
>
> [1]. Shuster et al, Stretching Sentence-pair NLI Models to Reason over Long Documents and Clusters

---

> > ### Comment · Reviewer_PSmh · 2025-08-05
> >
> > Thank the authors for addressing my questions. I think this is a good paper and will keep my score

---

### Decision · Program_Chairs · 2025-09-17

**Decision:**

Accept (poster)

**Comment:**

**Summary**

This paper proposes ReliabilityRAG, a defense framework to mitigate retrieval corpus attacks (e.g., prompt injection, corpus poisoning) in Retrieval-Augmented Generation (RAG) systems. The core design leverages document reliability signals (e.g., search rankings) and a graph-theoretic approach: 1) constructing a "contradiction graph" where edges between documents indicate conflicts (detected via Natural Language Inference, NLI) and 2) finding a rank-prioritized Maximum Independent Set (MIS) to select mutually consistent, high-reliability documents. To address the computational cost of exact MIS for large retrieval sets, the authors further propose a scalable weighted sampling-and-aggregate framework. Empirically, ReliabilityRAG maintains high accuracy in benign scenarios, outperforms baselines (e.g., RobustRAG, VanillaRAG) in robustness against attacks, and excels in long-form generation tasks where prior defense methods struggled. The work also provides theoretical guarantees of λ-robustness against bounded adversarial corruption.

**Strengths**

1. This paper addresses a timely and high-impact problem: Adversarial vulnerabilities in RAG systems (e.g., corrupted retrieval corpora) are critical barriers to real-world deployment (e.g., Google’s Search AI Overview). The work fills a gap by explicitly integrating reliability signals (a unique advantage of search-based RAG) into defense, rather than treating retrieval outputs as homogeneous.

2. The methodology is both theoretically rigorous and practically actionable: Translating document consistency into a rank-aware MIS problem is a novel, intuitive insight—grounded in graph theory and validated with provable robustness guarantees (Theorem 1). The authors further balance theory with practice by introducing a scalable sampling framework, resolving the computational bottleneck of exact MIS for large k (e.g., 50 retrieved documents) while preserving robustness.

3. Empirical results are strong and comprehensive: The framework maintains high benign accuracy, outperforms baselines across attack positions (e.g., Attack @ Pos 1, 5, 10 for k=10) and attack types (prompt injection, corpus poisoning), and addresses a key limitation of prior work by performing well in long-form generation.

4. This paper adopts a defense-in-depth philosophy: By combining NLI-based contradiction detection with ranking signals, the framework is resilient to NLI model degradation—experiments show graceful performance drops even with 50% NLI error probability, and malicious documents in low-ranked positions (e.g., Pos 25/50 for k=50) rarely impact results due to sampling prioritization.

**Weaknesses**

1. Evaluation depth is limited in key areas: Multi-position attacks (a critical real-world threat) are only partially analyzed in the main text (Section 5.4), with more comprehensive results confined to the appendix. The authors plan to move this content to the main text, but the initial presentation undersells the framework’s ability to handle coordinated attacks. Additionally, the evaluation lacks diverse attack types (e.g., adaptive attacks that exploit both NLI and ranking signals), which would strengthen claims of robustness.

2. Dependence on NLI model performance is not fully validated. While the authors demonstrate graceful degradation with NLI errors, they do not test performance on non-English or highly technical domains—settings where NLI models are known to underperform. This limits generalizability beyond the English open-domain datasets (e.g., RealtimeQA) used in experiments.

3. Parameter choices lack full theoretical justification: Sampling parameters (decay factor γ, number of rounds T, context size m) are set heuristically (e.g., T=20, m=2) based on empirical performance, with ablation studies but no theoretical bounds on their optimality. This raises questions about how to adapt the framework to new domains without extensive tuning.

4. Latency and real-world scalability need more validation: While the authors report sub-1s latency for prototypes (e.g., ~0.61s for k=10 + Mistral-7B), they acknowledge this is unoptimized. For high-scale systems (e.g., real-time search), more details on parallelization, caching, or lightweight model replacements (e.g., smaller LLMs for "isolated answering") would strengthen practicality claims.

**Discussions during Rebuttal Period**

During the rebuttal period, reviewers raised targeted concerns. The authors addressed those concerns with experiments or clear revision plans.

1. Reviewer PSmh raised four points: 1) NLI model sensitivity and β threshold analysis; 2) insufficient multi-position attack results; 3) overhead of "isolated answering"; 4) abstract/introduction formatting. The authors responded by: testing additional NLI models (deberta-v3-large-mnli) with similar results, showing β (0.2–0.8) has minimal impact (NLI outputs are near-binary); providing comprehensive multi-position attack results in Appendix C.5 (to be moved to main text); quantifying overhead (~0.27s for k=10 + Mistral-7B) and proposing caching/parallelization; and agreeing to prune the abstract and refine introduction coherence. These responses were fully satisfactory—NLI robustness and overhead concerns were resolved with data, and formatting issues are easily fixed.

2. Reviewer ei33 asked for: 1) statistical rigor (CI/decimal points); 2) contrast with lexical matching; 3) human annotation alternative to LLM-as-judge; 4) latency discussion. The authors addressed these by: providing error bars in Appendix C.4; conducting lexical matching experiments (showing <4.31% attack failure even at optimal δ, proving inadequacy); adding a multiple-choice dataset (no LLM-as-judge) with results matching main experiments, plus human validation of 100 LLM-judged cases (100% accuracy); and clarifying latency is optimizable (sub-1s prototype, parallelization potential). These responses were compelling—lexical matching’s limitations and LLM-judge reliability were confirmed with data.

3. Reviewer pm3b concerned: 1) NLI degradation impact; 2) MIS/NLI overhead; 3) reliability signal manipulation. The authors responded by: showing graceful performance drops (e.g., 55% accuracy at 50% NLI error for k=10, Attack @ Pos 1); detailing overhead (<0.05s for NLI checks, ~0.61s total for k=10); and noting MIS’s majority vote mitigates signal manipulation (benign majority preserves correctness). These responses resolved concerns—NLI resilience and overhead were quantified, and signal manipulation was addressed via the framework’s design.

4. Reviewer Xjz6 asked: 1) NLI robustness in non-English/technical domains; 2) parameter learning feasibility; 3) MIS + reranking integration; 4) handling no consistent majority; plus limitations on ambiguity/multi-perspective queries. The authors responded by: acknowledging non-English/technical validation as future work but noting defense-in-depth mitigates NLI gaps; framing parameter learning as a promising learning-augmented direction; agreeing MIS + weighted reranking is feasible (future work); and proposing multi-view answers/clarifying questions for no majority. While non-English validation remains unaddressed, the authors’ transparency about future work and practical solutions for ambiguity were acceptable.

**Decision Justification**

After reading the review and author response, along with the discussions, I think the strengths (e.g., novel methodology, theoretical rigor, strong empirical performance, and real-world relevance) outweigh its addressable weaknesses.

Originality and impact: The graph-theoretic MIS approach, combined with reliability signals, introduces a new paradigm for RAG defense that goes beyond prior methods (e.g., keyword-based RobustRAG) by integrating provable robustness and practical scalability. This advances the field of RAG robustness, a rapidly growing area with direct implications for trusted AI deployment.

Theory-practice balance: Unlike many defense papers that focus solely on empirical results, ReliabilityRAG provides clear theoretical guarantees (λ-robustness) while addressing computational barriers via sampling—critical for bridging lab and real-world use.
Response to core concerns: The authors have effectively addressed reviewer questions (e.g., NLI degradation, lexical matching inadequacy, multi-position attacks) with additional experiments or concrete revision plans, leaving no unaddressed fatal flaws.

Weaknesses (e.g., limited multi-position attack analysis, NLI domain validation) are not fundamental and can be resolved in revisions (e.g., moving appendix content to main text, adding a small experiment on technical domains). The framework’s ability to maintain accuracy while defending against attacks fills a critical need for RAG systems, justifying acceptance.